# Increasing CB2 Receptor Activity after Early Life Stress Prevents Depressive Behavior in Female Rats

**DOI:** 10.3390/biom14040464

**Published:** 2024-04-10

**Authors:** Susan L. Andersen

**Affiliations:** Boston Children’s Hospital, Harvard Medical School, Boston, MA 02115, USA; susan.andersen@childrens.harvard.edu

**Keywords:** adversity, CB2, depression, female, inflammation, microglia, PV, stress

## Abstract

Early adversity, the loss of the inhibitory GABAergic interneuron parvalbumin, and elevated neuroinflammation are associated with depression. Individuals with a maltreatment history initiate medicinal cannabis use earlier in life than non-maltreated individuals, suggesting self-medication. Female rats underwent maternal separation (MS) between 2 and 20 days of age to model early adversity or served as colony controls. The prelimbic cortex and behavior were examined to determine whether MS alters the cannabinoid receptor 2 (CB2), which has anti-inflammatory properties. A reduction in the CB2-associated regulatory enzyme MARCH7 leading to increased NLRP3 was observed with Western immunoblots in MS females. Immunohistochemistry with stereology quantified numbers of parvalbumin-immunoreactive cells and CB2 at 25, 40, and 100 days of age, revealing that the CB2 receptor associated with PV neurons initially increases at P25 and subsequently decreases by P40 in MS animals, with no change in controls. Confocal and triple-label microscopy suggest colocalization of these CB2 receptors to microglia wrapped around the parvalbumin neuron. Depressive-like behavior in MS animals was elevated at P40 and reduced with the CB2 agonist HU-308 or a CB2-overexpressing lentivirus microinjected into the prelimbic cortex. These results suggest that increasing CB2 expression by P40 in the prelimbic cortex prevents depressive behavior in MS female rats.

## 1. Introduction

Exposure to early life adversity is associated with depression that emerges during adolescence [1,2]. The delayed emergence in behavioral symptoms suggests that stress exposure interacts with maturational processes, although little is known about the process [3]. For example, synaptophysin changes, a synapse marker, are not observed until the overproduction of synapses during adolescence fails to occur in animal models of adversity [4]. In humans, brain morphometry changes in maltreated individuals appear after childhood [3,5,6]. Part of the delayed effects may be due to compensatory reactions initiated following stress exposure that eventually fail with further maturation. Neuroinflammation models suggest that early stress increases inflammatory processes that predispose the individual to depression [7,8]. Observed cytokine levels released from microglia in animals exposed to early-life stress accompany neuronal loss [9].

Microglia are essential during brain development by maintaining homeostasis and aiding in the normal pruning processes during childhood and adolescence [10,11,12]. Microglia are also known for their involvement in inflammation [13]. Exposure to stress increases inflammatory processes at all ages. However, when exposure to adversity occurs during development, stress further primes microglia to later challenges to produce an enhanced response ([12,14]; reviewed by [15]). Elevated cytokine release, the mediators of inflammation, accompany microglia priming. Both the cytokines interleukin (IL)-1b and IL-6 are elevated following maltreatment in humans [16] and in animal models, including maternal separation [17,18,19]. The cytokine IL-6 is associated with depressive-like behaviors that appear during adolescence in humans [20,21].

In addition to inflammation, adults with mood disorders have reduced cortical GABA levels [22,23,24]. Adolescent GABA levels negatively correlate with anhedonic symptoms of major depression [25]. Changes in the GABAergic interneuron that expresses the calcium-binding protein of parvalbumin (PV) are critical in developing cortical circuitry [26,27]. Exposure to maternal separation reduces PV levels relative to control rats during adolescence [17,28,29,30]. The cytokine IL-6 negatively affects PV neurons [31], so, unsurprisingly, both cortical PV and peripheral levels of IL-6 correlate with learned helplessness behavior in maternally separated rats [30], further supporting their role in depression.

Reducing inflammation may provide a preventative intervention for individuals exposed to early adversity. Cyclooxygenase-2 (COX-2) is part of the inflammatory pathway associated with increased cytokine release. Levels of COX-2 are elevated following maternal separation in adolescent rats [17]. Treatment with a COX-2 inhibitor or central administration of the anti-inflammatory cytokine IL-10 before adolescence prevents both PV loss [17] and depressive behavior [32]. However, COX-2 inhibitors are unsuitable for long-term treatment due to untoward side effects [33]. The type 2 cannabinoid receptor (CB2) regulates the COX-2 pathway [34] and decreases inflammation [35] and thus may offer another treatment approach for depression [36].

In support of the role of the CB2 receptor for treatment, recent evidence shows that individuals with a maltreatment history use more ‘medicinal’ marijuana than non-abused individuals. Populations exposed to early adversity/sexual abuse initiate marijuana use earlier in life compared to a non-abused population [37]. In the periphery, activation of the CB2 receptor increases macrophage autophagy that inhibits nucleotide-binding and oligomerization domain (Nod)-like receptor family pyrin domain-containing 3 (NLRP3) initiation and activation, decreasing the inflammatory cascade involving IL-1β [38,39,40]. Elevated levels of IL-1β following MS in females are found [41], but not consistently [19]. IL-1β is associated with anhedonia [42] and can induce the formation of IL-6 [43].

Until recently, scientists believed CB2 receptors were located almost exclusively in the periphery [44,45]. Newer data show that the CB2 receptor is upregulated in response to pain and injury in the brain [46], raising the possibility that the receptor is involved in the effects of early life stress. Research in other models of adversity (e.g., hypoxia and traumatic brain injury) shows that activated microglia increase CB2 receptor expression [47,48,49,50]. However, microglia numbers identified with iba-1 are not upregulated in the prefrontal cortex of maternally stressed animals but show morphological changes consistent with activation [51]. However, elevated microglia activity (i.e., phagocytosis) is observed in other brain regions, including the hippocampus [38]. Studies thus far have identified CB2 on microglia [52,53], astrocytes [54], and neurons [55], leading us to investigate whether changes in CB2 activity mediate neuroinflammatory changes in the prefrontal cortex.

The following studies establish the inter-relationships between the maternal separation model of adversity, PV cell loss, neuroinflammation, and CB2 treatment. First, Western immunoblot was used to determine whether exposure to early adversity alters signaling pathways associated with inflammatory pathways, including those linked to CB2 receptors on microglia. We measured the E3 ubiquitin enzyme MARCH7 and its target, [NLRP3] [53]. Notably, NLRP3 is found in microglia, astrocytes, and neurons [56]. Second, immunohistochemistry characterized the effects of maternal separation on PV and CB2 receptors across ages. The focus on the trajectory between P25, 40, and 100 reflects childhood, adolescence, and adulthood in rats. More importantly, research shows that the depressive effects associated with adversity manifest during adolescence in humans and rats [3]. Further localization to microglia was also assessed with additional immunohistochemical analysis. Third, the location of CB2 receptors affecting PV neurons was determined behaviorally and within the microcircuitry of the prelimbic cortex by examining depressive-like behaviors through a CB2 receptor overexpressing virus.

While both sexes demonstrate elevated inflammation, PV cell loss, and depressive behavior [17,28,30], the study focused exclusively on females. First, the effects of a COX-2 inhibitor prevent PV loss in both sexes, but its ability to prevent depressive effects has only been demonstrated in females [32]. Second, maternal separation increases microglia soma size and microglia arborization in females but not in male rats [51]. The results show that the CB2 receptor interacts with PV expression and decreases depressive behavior in maternally separated females, although the cellular location of this effect remains unclear.

## 2. Material and Methods

### 2.1. Subjects

Pregnant female multiparous Sprague-Dawley rats (250–275 g) were obtained from Charles River Laboratories (Wilmington, MA, USA) on day 16 of gestation. The day of birth was designated as postnatal day 0 (P0). One day after birth, all litters were culled to 10 pups (5 males and 5 females), and pups were randomly assigned to each group. Only one pup/sex per litter was assigned to a single condition. A total of ten different litters from each condition were used.

Litters were randomly assigned to an animal facility-reared control group (“Con”) that was undisturbed except for weekly cage cleaning, weighed once a week, or a second group that underwent maternal separation. Pups in the maternal separation group were isolated for 4 h/day between P2 and P20 at a thermoneutral temperature [4]. Separations occurred between the window of 7:00 AM–12:00 PM. All rats were housed with food and water available ad libitum in constant temperature and humidity conditions on a 12-h light/dark cycle (light period 7:00 AM–7:00 PM).

Rats were weaned on P21–22 and group-housed with same-sex littermates with 3–4 rats/cage until experimentation. Only one female/litter was used/age/condition to avoid litter effects. Female rats were tested during the juvenile stage (P25), adolescence (P40), or adulthood (P100) as defined by milestones [57]. These experiments were conducted following the 1996 Guide for the Care and Use of Laboratory Animals (National Institutes of Health) and were approved by the Institutional Animal Care and Use Committee at McLean Hospital.

### 2.2. Study Design

The design of the study is shown in Figure 1. The prelimbic cortex was selected for analysis due to its role in depression and modulation of the hypothalamic–pituitary axis activity [58,59,60]. In addition, PV changes in maternally separated animals were only observed in the prefrontal cortex, not the hippocampus [17].

### 2.3. Experiment 1: Western Immunoblot of Pathway Signaling

Western immunoblots determined protein levels within an inflammation signaling pathway. Samples from the prelimbic prefrontal cortex from n = 5 Control and n = 5–6 maternally separated females at P40 were obtained and rapidly frozen. Tissue was homogenized in RIPA lysis buffer (Cell Signaling Technology, Boston, MA, USA), and the individual protein concentration was quantified using a Bio-Rad Protein Assay (Bio-Rad Laboratories, Hercules, CA, USA). Prepared protein samples (20 µg protein) were loaded into individual wells with a Bis-Tris-4-12% NuPage gel, with all conditions represented in a single blot. Samples were separated at 200 V using a NuPage MOPS running buffer with NuPage antioxidant present (Invitrogen, Waltham, MA, USA). Separated protein was transferred to an FL-PVDF Membrane, pore size = 0.45 µm, with NuPage Transfer buffer and 10% ethanol at 30 V constant for one hour. Membranes were blocked for 1 h with Odyssey Blocking buffer (Invitrogen) before incubating with primary antibodies. Primary antibodies were diluted in Odyssey blocking buffer/PBS (50% vol/50% vol) at a concentration of anti-MARCH7 (1:500; mouse; Millipore, Burlington, MA, USA) and its control VCP (1:10 K: rabbit; Sigma, Saint Louis, MO, USA) or NLRP3 (1:200; mouse; AbCam, Waltham, MA, USA) and actin (1:10,000; Sigma) as the control for the second immunoblot. Membranes were washed four times for 5 min each with 0.01 M PBS + 0.1% Tween-20 before incubating in secondary antibodies (LiCor Biosciences, Lincoln, NE, USA) for one hour. Li-Cor secondary antibodies were IRDye 800CW donkey anti-mouse (IgG; H&L) and IRDye 680RD goat anti-rabbit (IgG) (H&L). Membranes were washed to remove excess antibodies before imaging on a LiCor Odyssey DLX, and each band’s optical density and size were quantified with Li-Cor Odyssey Image System (LI-COR Biosciences, Lincoln, NE, USA). The size of the individual control band corrected samples for differences in protein loading.

### 2.4. Experiment 2: Anatomical Changes in PV and CB2 and Microglia

The first cohort of subjects (n = 4/group) were deeply anesthetized at P25 (juvenile), 40 (adolescence), and 100 (adulthood), transcardially perfused with phosphate-buffered saline (PBS; pH = 7.4) followed by 4% paraformaldehyde in PBS [17]. The brains were then post-fixed for 4 h at 20 °C and equilibrated in a 30% sucrose solution. The brains were sliced (40 μm) coronally, and a total of four sections 240 µm apart were analyzed. Double labeling of CB2 and PV immunoreactive neurons was quantified with unbiased stereology as a function of age and maternal separation (described below). 

A second cohort of animals, consisting of two maternally separated females at P40, was used for confocal microscopy. The brains were processed similarly to the abovementioned methods, but their brains were sliced at 200 µm and stained for immunofluorescence for PV and CB2. Images were taken with a Leica confocal microscope at 100× with oil submersion. For the third immunohistochemistry analysis, 40 µm sections from the first cohort (P40) were used for the triple label, descriptive analysis of CB2, PV, and microglia.

#### 2.4.1. Immunohistochemistry and Stereology of PV and CB2 Receptors

At P25, P40, or P100, rats were deeply anesthetized and intracardially perfused. The tissue was processed using our standard immunohistochemical methods [17]. Forty-micron brain sections were washed in TritonX-100 PBS before and after labeling. Sections double labeled with PV (1:10,000; Sigma monoclonal made in mouse) and CB2 (1:50; Cayman, Ann Arbor, MI, USA; polyclonal antibody made in rabbit) overnight, enhanced with secondary (biotinylated anti-mouse and anti-rabbit secondary serums [both 1:500; Sigma, respectively), and streptavidin (1:4000; Invitrogen, Camarillo, CA, USA). Staining was visualized with diaminobenzidine and diaminobenzidine + nickel for PV and CB2, respectively, in H_2_O_2_. All steps were preceded and followed by washes in PBS with Triton X-100. Sections were mounted on gelatin-coated slides, dehydrated, and cover-slipped with Permount (Thermo Fisher Scientific, Waltham, MA, USA). 

The density of PV and CB2/PV double-labeled cells was estimated with the Stereo Investigator Image Analysis System (MBF BioScience, Williston, VT, USA) on a Zeiss AxioImage2 microscope. Four serial coronal sections (intersection interval 240 µm)/animal of both hemispheres were analyzed, and the average value was used in subsequent calculations to represent each subject. The entire prelimbic cortex was outlined at 4× magnification, and the number of immunoreactive PV and CB2/PV cells was counted at 20×. Tracings of the prelimbic cortex boundaries were used to calculate each section’s surface area (a). While double-labeled cells were readily identified, CB2 immunoreactivity alone was unreliable and was not performed. The investigator was strictly blinded to the conditions for all analyses.

Estimates of total cell density were based on the surface area (a) in each section. All counting was conducted by the investigator who was blind to the condition of the tissue. The density of immunoreactive cells (cells/mm^2^) was based on the total number of PV and CB2/PV cells divided by the sum of areas from all outlined regions. The total volume of the prelimbic cortex was calculated according to the Cavalieri principle [61] as *v* = *z* × *i* × *Sa*; *z* is the thickness of the section (40 µm), and *i* is the section interval (*i* = 24; i.e., number of serial sections between each section and the following one within a compartment).

#### 2.4.2. Immunohistochemistry PV, CB2 Receptors, and iba-1 as a Marker of Microglia

A triple-label immunohistochemistry assessment was performed once it was determined that CB2 was not co-localized with PV. PV and CB2 were described above with fluorescent secondary antibodies (anti-mouse and anti-rabbit secondary serums). The sections were triple labeled with the addition of ionized calcium-binding adaptor molecule 1 (iba-1, a marker of microglia; 1:250; Wako Chemicals, Richmond, VA, USA, raised in goat). Staining was enhanced with an anti-goat secondary (AlexaFluor, LiCor, Lincoln, NE, USA; 660) and visualized at 660 λ on a Zeiss Axoscope2 microscope with Apotome extension for enhanced clarity. These sections were imaged only and not counted.

#### 2.4.3. Confocal Image of CB2 and PV (PV) Interactions

Confocal microscopy was used to determine whether PV and CB2 markers are co-localized. A Leica TCS-SP8 equipped with laser excitation wavelengths of 488 and 560 nm, a unique prism-based spectral dispersion system for maximizing image quality and contrast, and detectors for simultaneous multicolor imaging of immunofluorescent labels were used for this image.

Tissue was processed with our standard immunohistochemical methods described in 2.4.1 [17]. The brains were sliced (200 μm) and double-labeled with antibodies against PV (1:10,000; Sigma monoclonal raised in mouse) and CB2 (1:50; Cayman, Ann Arbor, MI, USA, raised in rabbit), and enhanced with fluorescent secondary AlexaFluor 488 and 568 antibodies (anti-mouse secondary serum [1:500; Sigma] and anti-rabbit secondary; 1:500). All steps were preceded and followed by washes in phosphate-buffered saline–Triton X-100. Sections were mounted on gelatin-coated slides, dried, and cover-slipped. Fluorescent images of PV and CB2 were taken with a Leica confocal microscope at 100× with oil submersion and visualized at 480 and 560 λ.

### 2.5. Manipulation of CB2 Receptor Activity

Research shows that CB2 receptors in the brain are expressed under conditions of direct damage or stress [47,48,49,50]. Changes in CB2-related behaviors were examined only in the maternally separated females, as no changes were observed in control subjects.

#### 2.5.1. Treatment with the CB2 Agonist, HU-308

To determine whether depressive-like behavior could be prevented, female rats that underwent maternal separation were treated with the dimethyl sulfoxide vehicle (DMSO; n = 7) or the CB2 agonist, HU-308 in DMSO (2.5 mg/kg; n = 6) on P30, 32, 34, 36, and 38 with testing on P40. Subjects received i.p injections at 7 a.m. These were the same ages and methods used for COX-2i reversal of PV loss and depressive behavior in maternally separated rats [17,30]; this approach also reduces the effects of DMSO as a mild irritant. The 2.5 mg/kg dose of HU-308 is based on a compromise between ineffective and effective doses in the literature. Lower doses of 0.5 and 1 mg/kg HU-308 were ineffective in animal models of seizures [62]. A 5 mg/kg dose effectively prevented microglia proliferation in a viral encephalitis model [63].

#### 2.5.2. CB2 Receptor Overexpressing Lentiviral Vector

Lentiviral vectors that overexpressed either EGFP (pLV[Exp]-Puro- [64] > EGFP) or CNR2 (CB2 receptor plus an EGFP reporter; pLV[Exp]-Puro- [64] > hCNR2[NM_001841.3]/EGFP) via the CD11b promotor pGEM3zf(-) that was produced commercially on a Promega backbone (VectorBuilder). Briefly, the CD11b promoter in pGEM3zf(-) was a gift from Daniel Tenen (Addgene plasmid no. 26168; http://n2t.net/addgene:26168; RRID:Addgene_26168) [64]. The GenBank accession number is M84477 M76724 M80772. The CB2-expressing viral vector has an EGFP reporter that is visible when the virus is actively expressed. The publicly available ImageJ v.1.54i; (http://imagej.nih.gov/ij; accessed on 24 February 2024) software was used to quantify the amount of CB2-EGFP overexpression by measuring the amount of immunofluorescence in the prelimbic region where the virus was expressed. An n = 3 subjects/group were analyzed one day, one week, and two weeks after the virus was injected.

Following our previous methods [65], female rats (n = 7/group) that underwent maternal separation were anesthetized with a ketamine/xylazine mixture (80/12 mg/kg, respectively) at P33. Subjects received 0.5 μL of virus (>10^9^ transducing units per μL) bilaterally into the prelimbic cortex at stereotaxic coordinates of AP ± 3.8, ML ± 0.6, DV ± 1.6; [66]. Seven days after surgery, behavioral testing began to allow for viral expression [67]. Expression was stable throughout all experiments, and placement was confirmed by histology. Only subjects where the virus was in the prelimbic cortex were used. While the intended target was microglia via the CD11b promotor, in the absence of immunohistochemistry directly localizing CB2 overexpression to microglia, the manipulation will be identified as CB2-EGFP overexpression only.

### 2.6. Behavioral Measure of Depression

A preliminary investigation of whether increased CB2 activity could reduce depressive behavior was determined in two ways:(a)Learned helplessness test following CB2 agonist treatment with HU-308 (2.5 mg/kg between 30–38 days of age) or following prelimbic expression of a lentiviral vector that selectively overexpresses receptors (CB2-EGFP).(b)Sucrose preference test (a measure of anhedonia) following the HU-308 agonist or the CB2-EGFP viral vector.

#### 2.6.1. Sucrose Preference Test

To measure reduced sensitivity to natural rewards, the sucrose preference test was used to compare Controls and maternal separation animals in subjects beginning at P41. A dose-response assessment of sucrose preference consisted of a baseline (0%; water) 0.25%, 0.5%, and 1% sucrose concentration for the CB2-EGFP virus (n = 8/group), but only at 1% for HU-308 (n = 6). The baseline was measured across two days of water on the right and left sides to ensure no side bias existed (50 ± 10% drinking from either side). The placement of sucrose-containing bottles was counterbalanced with water bottles across for each of the two days they were presented. The amounts of water and sucrose consumed in 24 h were the dependent variable and were measured at 7:00 AM for each experimental day. The amount of sucrose consumed was expressed as follows:% sucrose = 100% × sucrose/water

The average across the two days of drinking was used in the statistical calculations.

#### 2.6.2. Learned Helplessness

Rats were placed on one side of a shuttle box (MedAssociates, Fairfax, VT, USA) and tested for depressive-like behavior following our established protocols [30,68]. Rats could terminate a 1-mA foot shock by shuttling to the other side for trials 1–5 or shuttling to the other side and back again for trials 6–30. This response was cued by a tone that preceded the shock by 2 s. The shock remained on for 30 s or until terminated by the appropriate behavioral response. Consistent with previous studies [68], data from the first five trials were not used in the analyses as subjects were learning the appropriate behavioral response. The number of escape failures and the mean latency to escape were reported for trials 6–30. 

### 2.7. Statistical Analyses

Statistical analyses were conducted using SPSS (v26; IBM, Cambridge, MA, USA) with a significance set at *p* < 0.05. An ANOVA was used to compare Western immunoblot protein levels in Experiment 1 and depressive behavior in Experiment 3. An Age × Condition ANOVA assessed differences in PV and CB2/PV immunoreactivity in Experiment 2, whereas the confocal study is descriptive. Finally, a repeated measure ANOVA, with sucrose concentration as the repeated measure, by Condition, was used to analyze sucrose preferences in Experiment 3.2.2. 

## 3. Results

### 3.1. Experiment 1: Western Immunoblot of Pathway Signaling

Figure 2a (modified from [69]) illustrates how the CB2 receptor modulates inflammation by regulating the MARCH7 enzyme, which regulates NLRP3 expression [70]. Each protein from both conditions was corrected by its control (actin or VCP) and then expressed as a percentage of the control group’s average. These values were analyzed with a one-way ANOVA with the condition as the between-subjects variable. The analysis revealed that MARCH7 was reduced by 20% in maternally separated subjects relative to control subjects (F1,9 = 5.0, *p* < 0.05; Figure 2b and Appendix A). In contrast, NLRP3 was increased by 100% due to disinhibition (F1,9 = 5.6, *p* < 0.05; Figure 2c). A second cohort replicated these findings.

### 3.2. Experiment 2: Anatomical Changes in PV and CB2

#### 3.2.1. Experiment 2.1: Immunohistochemistry of CB2 and PV Neurons

Single (PV) and double (CB2/PV) label immunohistochemistry (Figure 3a,b) was analyzed across age (P25, 40, and 100) and condition (Con/MS) using unbiased stereology and the number of immunoreactive cells/mm^2^ were tabulated. The cell/mm^2^ number served as the dependent variable, with the age and condition serving as between-subjects variables and analyzed with a two-way ANOVA. While CB2/PV staining is readily observable and thus quantifiable, due to faint staining of CB2 alone, quantification of this single maker could not be performed reliably (Figure 3c). 

In contrast to our previous studies in male rats [17,30], a significant main effect demonstrated increased PV immunoreactive cells in maternally separated females relative to controls (F1,17 = 4.7, *p* < 0.05), with no significant interaction (Figure 3d). The total amount of PV-immunoreactive cells was marginally significant (*p* = 0.05; Figure 3e) as was the interaction by age (*p* = 0.06). Double-labeled neurons of CB2/PV were counted (Figure 3f). While CB2/PV did not appreciably change with age in control subjects, CB2/PV declined across age in the maternally separated animals (Age × Condition interaction: F2,17 = 4.20, *p* < 0.05). When the interaction was dissected further with post-hoc ANOVA, it was revealed that CB2/PV neurons significantly changed across age in the MS group, with significant differences observed between P25 and P100 (*p* < 0.05) but counts were not different between P25 and P40 (*p* = 0.07). CB2/PV immunoreactivity did not change significantly across age in the Con group.

#### 3.2.2. Experiment 2.2: Confocal Image of CB2 and PV (PV) Interactions

The enhanced resolution of confocal microscopy was used to determine whether PV and CB2 markers are co-localized (Figure 3g). These results clarify the results of 3.2.1 by showing that the CB2/PV neurons are PV neurons surrounded by CB2-expressing cells that are likely microglia.

#### 3.2.3. Experiment 2.3: Triple Labeling of CB2, PV, and the Microglia Marker iba-1

Figure 4 shows the triple-label immunohistochemistry of CB2, PV, and iba-1, a marker for microglia. Yellow color in the image is indicative of CB2 and possible PV neurons, however, the addition of the pink color in the image suggests that all three markers are located on the same neuron. However, when this image is placed in the context of Figure 3g, the CB2/iba-1 are likely co-localized and wrapped around the PV neurons, as suggested by the close relationship of CB2-immunoreactive neurons with a single labeled PV neuron.

### 3.3. Experiment 3: Behavioral Measure of Depression

#### 3.3.1. Experiment 3.1: Treatment with the CB2 Agonist, HU-308

To determine whether activation of CB2 signaling could prevent the depression-like behavior in MS females, subjects were treated with the CB2 agonist 2.5 mg/kg HU-308 on P30, 32, 34, 36, and 38 following our previous intervention protocol (timeline in Figure 5a; [17]). The treatment was partially effective in reducing the depressive-like behaviors of anhedonia by P40 as assessed by escape behavior and sucrose preferences (Figure 5b–d).

Preferences for 1% sucrose between the vehicle and HU-308 groups were non-significant (*p* > 0.4; Figure 5b). The latency to escape or the total number of escape failures was analyzed from trials 6–30, as earlier research shows the animals are still learning the protocol during the first five trials [68]. The CB2 agonist HU-308 had no significant effect on the latency to escape (*p* > 0.1) but reduced the number of escape failures (F1,11 = 25.3, *p* < 0.005) relative to vehicle-treated females.

#### 3.3.2. Experiment 3.2: Treatment with a Lentiviral Vector That Expresses CB2 Receptors

The prelimbic cortex (Figure 6a) of maternally separated females was injected with a lentiviral vector that overexpressed the CB2-EGFP receptor or a control (EGFP) virus (Figure 6b,c), and anhedonic behavior was determined. The amount of immunofluorescence produced by the EGFP reporter in the CB2-EGFP virus significantly increased 2.3-fold after one week of surgery and 4-fold after two weeks (F2,6 = 66.1, *p* < 0.001; Figure 6d). A repeated measures ANOVA with the concentration of sucrose (0, 0.25, 0.5, and 1%) as the repeated measure and virus condition as a between-subjects factor was significant (F1,14 = 4.6, *p* < 0.05; Figure 6e). Figure 6e shows that sucrose preferences are higher in the maternal-separated animals treated with the control virus than those treated with the CB2-EGFP virus, suggesting less anhedonia.

Both the number of escape failures and latency to escape were reduced by the CB2-EGFP virus relative to controls (Figure 6f,g; ANOVA F1,14 = 12.7 and 18.6, *p*’s < 0.005).

## 4. Discussion

The results of the current studies demonstrate a role for the CB2 receptor in depressive-like behavior in adolescent female rats following exposure to early adversity; additional studies are needed to show localization to microglia conclusively. Initial increases in CB2 receptor-associated signaling molecules followed by their compensatory changes provide a possible mechanism for the elevation of cytokine production in maternally separated females. Notably, MARCH7, a regulatory enzyme for NLRP3 activity, is reduced, whereas NLRP3, a critical factor in inflammasome activity, is elevated in the prelimbic cortex of maternally separated adolescent females. A single report shows elevated NLRP3 in the hippocampus of maternally separated mice [71], and the current findings now include NLRP3 alterations in the prelimbic cortex. Inflammasome activity is associated with danger signals and IL-1β production typically via TLR-4 receptors [72,73]. Additionally, CB2 receptors can modulate the inflammasome [74], thus serving as a potential target to reduce cytokine activity.

Consistent with the inflammasome as a therapeutic target, Wu and colleagues demonstrated that brief separations from the dam promoted resilient behavior and decreased NLRP3 [75]. Together, these two indices provide a pathway that potentially links the CB2 receptor to enhanced cytokine production during early adversity. NLRP3 is found on other cells, including epithelial cells, astrocytes, and some neuronal populations [56]. While the data are consistent with the proposed model in Figure 2, additional studies that directly manipulate the CB2 receptor on microglia and other cell types are needed to confirm this relationship with MARCH7/NLRP3. The role of astrocytes in this process should also be considered, as they also play a role in development and PV activity [76]. Regardless, the role of NLRP3 in inflammation and depression (reviewed in [56]) warrants further investigation in MS animals.

Reduced PV-immunoreactive neurons following maternal separation are associated with depressive behavior [30], and the current results enhance our understanding of this process. Here, single-labeled PV neurons declined across 22.1% between P25 and 40 in maternally separated animals, consistent with our previous findings of age-related loss [17,30]. However, PV levels are slightly higher in MS females than in controls, which has also been reported [17,30]. The current study extends these findings to show that the delayed depression observed in the maternal separation model is further related to a progressive decrease in CB2/PV expression across age. The initial increase in CB2 expression associated with PV neurons at P25 likely represents a compensatory mechanism of primed that fails over time. By P40, CB2-associated rescue of these PV neurons (i.e., CB2/PV expression) is insufficient to prevent depressive behavior. Treatment with a CB2 agonist before P40 reduced depressive behavior (escape fails but not sucrose preferences). Moreover, the CB2-EGFP lentiviral vector confirmed the role of overexpression of CB2 in depressive effects (both escape fails and enhanced sucrose preferences).

Whether these changes in CB2 expression are directly related to microglial activity remains to be determined. The iba-1 marker used in Experiment 2.2 reflects microglia but not necessarily those in the activated state [77]. Colocalization of iba-1 and CB2 is suggested by immunohistochemistry, but the close proximity of the three markers (iba-1, PV, and CB2) is difficult to parse, even with a high-resolution microscope. Alternative explanations include the possibility that CB2 *is* expressed on PV (but see Figure 3g) and on iba-1 immunoreactive cells. Changes in iba-1 are found in adversity models. Some have found an upregulation of iba-1 following a shorter maternal separation (P0–10; [78]), whereas others using the limited bedding model of early adversity found reduced microglia ramification [79]. Similarly, Gildawie and colleagues did not observe a quantitative change using the same maternal separation paradigm as the current study (e.g., P2–20; [51]). Instead, these researchers observed a qualitative, morphological change in iba-1-immunoreactive microglia in juvenile females at the same age when CB2/PV cells were elevated; both studies found reduced effects by adolescence. Collectively, the close relationship between CB2/PV/iba-1 reflects stress-induced changes of inflammation and pruning that are known to occur [69].

While the intended target of the CB2-EGFP virus was microglia to parse out the CB2/PV/iba-1 issue, the immunohistochemistry in Figure 6 does not allow confirmed viral-mediated overexpression of CB2 specific to microglia. As a result, the conclusion about the role of microglia is speculative but still compelling. In other models of brain insults, including hypoxia-ischemia and middle cerebral artery occlusion [48], increased CB2 expression regulates enhanced or primed microglia activity to decrease inflammatory processes. Based on the current data, the elevation in CB2/PV counts observed at P25 in maternally separated animals suggests this protective mechanism is engaged early after the stress exposure. This proposed “protective” mechanism is consistent with the absence of depressive-like behaviors before adolescence (P40) in maternally separated animals (unpublished observations) or the loss of PV immunoreactivity in juvenile animals [17]. By P40, however, the decrease in CB2/PV counts, a down-regulation of MARCH7, and the upregulation of NLRP3 may indicate a failure of this process. Clearly, more work is needed to determine the time course and nature of these changes.

The observed increase in PV neurons at P25 is noteworthy, as it contradicts our earlier studies, where no change in PV number was observed at this age in male rats [17]. The likely explanation is that the counted PV neurons had enhanced staining due to the nickel associated with the CB2 label, thus allowing their detection that otherwise might not have been counted. Inspection of Figure 3b suggests that many of these PV neurons contain less PV protein than those observed in controls. Future studies will determine the density of the staining (or immunofluorescence if these secondary antibodies are used) to examine the amount of PV expression within each cell.

The results demonstrate that changes in the cannabinoid/inflammatory pathways are also evident in the prelimbic cortex and not only the hippocampus in stress paradigms. Hill and colleagues [80,81] and others [82,83] have established the importance of endocannabinoid signaling in adapting to stress, with many studies focusing on CB1-associated signaling in the hippocampus and other non-cortical regions. Maternal separation between P2–12 [84] or P1–15 [85] reduces the CB1 receptor in the cortex but minimally affects other endocannabinoids. The CB2 receptor was unchanged when examined in adulthood [85], consistent with our findings. The current study suggests that a more protracted maternal separation period, or later development, significantly affects CB2 receptors in the cortex. Consistent with the possibility of the influence of timing, depressive-like behavior (i.e., escape failures) was higher when the timing of the maternal separation paradigm occurred between P9–16 compared with P2–9 [86].

CB2 receptors may be modulated within a narrow developmental period. Cortical CB2 receptor mRNA increases following a brief period of maternal separation in males but not females [87]. The developmental profile of CB2 receptors in the cortex has yet to be determined, but the CB1 receptor shows peak cortical expression during juvenility and declines into adolescence [88]. Behavioral evidence from rats shows that the endocannabinoid system, and the CB2 receptor specifically, influences social development when assessed between 28 and 30 days of age [89]. Social interactions are another aspect of depression studied in animals. While not as specific as measuring CB2 receptors directly, indirect evidence for the role of CB2 in social interactions systemic co-administration of the CB1/CB2 agonist arachidonoylglycerol (2-AG) and a CB1 antagonist selects for CB2 [90]. This treatment should be biased toward CB2 activity and increased play behavior/social interactions. Rats that underwent maternal separation show reduced social interactions [91], supporting a loss of CB2 receptors in this model of adversity.

Elevated CB2 expression on microglia may regulate post-injury microglial activation and inflammatory functions. Indeed, future studies should investigate whether blocking the upregulation of CB2 receptors at P25 results in depressive-like behaviors and PV loss at this age. If so, a preventative treatment with a CB2 agonist could be developed to redirect the trajectory back onto a healthier path, as shown with a COX-2 inhibitor [17]. Clinical observations that CB2 agonists serve as an adjunct treatment for anxiety [92] could serve that purpose, as two-thirds of individuals with exposure to adversity experience anxiety that precedes their depression [3]. The current results offer new support for targeting CB2 receptors and associated signaling mechanisms in subjects exposed to early life stress as a preventative treatment.

However, caution is needed in interpreting the current results. Concern over commercially available CB2 antibodies has hampered research on this molecule [93]. The Cayman antibody binds to the intracellular 3rd loop (residues 228–242; [93]) of the N-terminus [94]. Testing of the Cayman antibody used in the current study has revealed high sensitivity to the CB2 receptor, as indicated by positive controls using mass spectrometry and blocking Western immunoblot expression with a CB2 antibody [95]. Dependence on mice knockouts to characterize CB2 antibodies is potentially flawed given different epitopes for antibody binding and knockout sites [93]. Until a full CB2 receptor knockout is available, relying on partial knockouts found in the Deltagen and Zimmer mice strain will continue. The Cayman antibody also binds to other proteins, but whether they are different epitopes of the CB2 receptor is still unclear [95]. Specificity for the CB2 receptor was reported to be at least 50–70%. Yet, the inflammatory signaling pathways and the behavioral studies are consistent with changes in CB2 activity following maternal separation. Most importantly, the behavioral effects of the CB2 agonist HU-308 are confirmed with the specific lentiviral vector expressing CB2-EGFP. Immunofluorescence of the reporter gene indicates viral overexpression is 2.3-fold higher than controls when the animals were behaviorally tested. However, additional confirmation with specific mRNA transcripts is needed and will provide greater insight into the degree of CB2 elevation found in maternally separated females.

Second, some of these studies are preliminary and provide a foundation for several lines of investigation. One limitation is that dose-response assessments of HU-308 are needed, and a narrowing of the ideal age to treat the abuse subjects for maximal effect. Other studies using this experimental compound have used higher doses (5 and 10 mg/kg) with success in changing physiological responses [63], but more behavioral assessments are needed. The need to establish whether a CB2 agonist would have the same effect in control subjects is equally important. CB2 receptor-mediated modulation of microglia is believed to occur only when the microglia are activated [48]. The effects of the CB2-EGFP virus suggest that introducing it into the prelimbic cortex was sufficient to increase CB2 expression following minimal physical trauma due to its delivery. However, localization to microglia was not confirmed, and therefore, additional studies will be needed to determine the cell-specificity of the observed behavioral effects. As such, the latency to escape and the number of escape fails are lower than those observed in the DMSO vehicle group for the CB2 agonist study.

The results of the current study demonstrate that selective activation of the CB2 receptor, and not the CB1 receptor, can be an effective trauma-relevant treatment [96]. A previous study shows that adolescent treatment with the CB1/2 agonist WIN55,212-2 in animals that underwent maternal separation (P7–14) showed significant improvement in short-term memory, spatial recognition, and social interaction tasks [97]. More to the point, the existing literature on adolescent tetrahydrocannabinol (THC)/CB1 exposure in animals suggests that a sensitive period occurs before adulthood, with most studies suggesting exposure before puberty onset in males (females were not examined) as critical [98,99]. However, the use of cannabis is not ideal as the phytocannabinoids affect multiple aspects of the endocannabinoid system (reviewed by [100]). Rodent studies in normal, non-stressed animals show that treatment with a CB1/2 agonist during early or mid-adolescence, but not in late adolescence or adulthood, alters GABA-A activity and disinhibits frequency-dependent activity in the prefrontal cortex to juvenile-like levels in adults that is consistent with an arrested development [101].

Several parallel findings between animals and humans following exposure to early life stress show the conservation of environmental programming on the brain [6]. More importantly, these parallels provide a bridge between species that facilitate the translation of novel findings in animals to humans and a means to test their impact. For example, changes in cortical GABA levels that are detectable using magnetic resonance spectroscopy in rats significantly correlate with Western immunoblot-quantified levels of PV [69]. Thus, magnetic resonance spectroscopy studies could examine whether GABA levels are decreasing in the brains of young cannabis users, especially those with a history of early maltreatment. If so, this vital biomarker, in conjunction with elevated inflammation markers, can be used to both identify those at risk for depression as well as likely candidates for a preventative intervention with a CB2 agonist.

## 5. Conclusions

The current studies support the protective role of CB2 receptors in the prelimbic cortex of maternally separated females. Specifically, early life stress elevates neuroinflammatory processes involving the inflammasome, which is associated with CB2 receptor signaling. Treatment with a CB2 agonist (or viral vector targeting to elevate prelimbic CB2 receptors) prevents the depressive phenotype in rats exposed to early life stress. Future studies are needed to determine whether chronic cannabis exposure sculpts the maltreated brain to prevent later affective illness or worsens depressive behavior during periods of withdrawal.

## Figures and Tables

**Figure 1 biomolecules-14-00464-f001:**
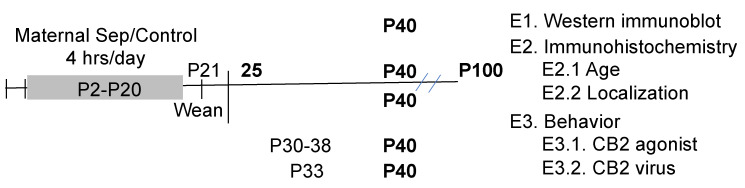
The timeline of the treatments and experimental design are shown. Ages in postnatal day (P) are shown with the ages of different manipulations identified. The bold indicates when the animals were assessed for behavior or immunohistochemistry.

**Figure 2 biomolecules-14-00464-f002:**
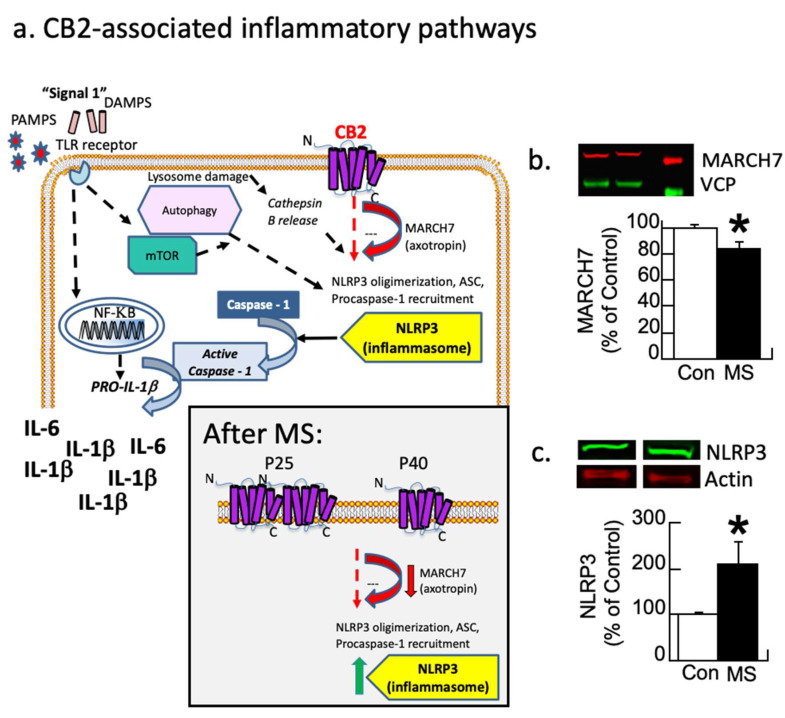
Inflammatory pathways disrupted by maternal separation (MS). (**a**) Schematic showing signaling pathways associated with inflammation. Most research focuses on the TLR pathway. CB2 receptor-associated pathways also modulate MARCH7/axotropin and NLRP3 as shown here. The gray box insert shows the observed changes following MS, where CB2 overexpression leads to a down-regulation followed by disinhibition of NLRP3. Modified figure from DOI: 10.1097/HRP.0000000000000325). (**b**,**c**) Female rats underwent MS or were controls (Con). At P40, the prelimbic prefrontal cortex was analyzed with Western immunoblots for MARCH7 and NLRP3 with VCP or actin as controls, respectively. (**b**) MS rats have a significant decrease (red arrow) in the regulatory enzyme MARCH7. (**c**) NLRP3 expression increases (green arrow) in MS females relative to controls (* *p* < 0.05). Means ± SE are presented (n = 6/5 for MARCH7 and 5/5 for NLRP3). A second cohort replicated this finding.

**Figure 3 biomolecules-14-00464-f003:**
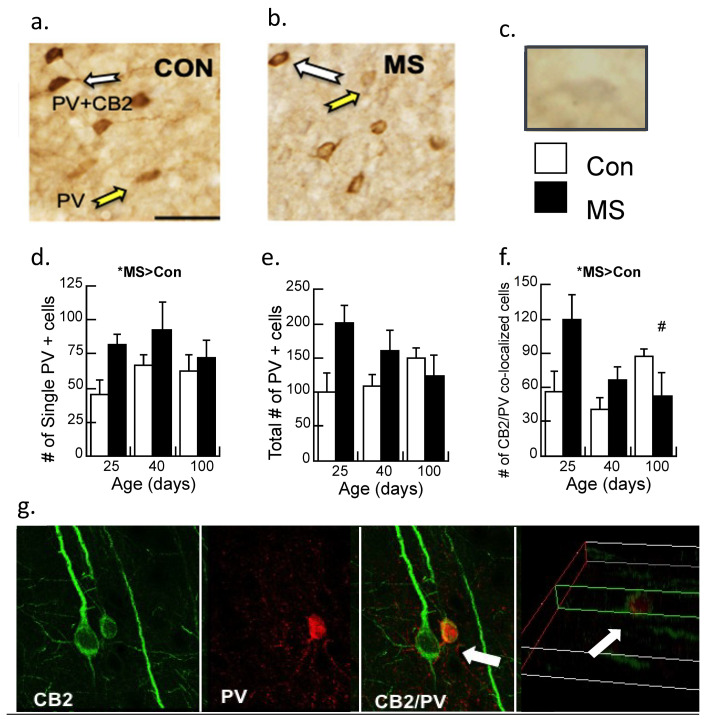
Immunohistochemistry of parvalbumin (PV) neurons and CB2 receptors in the prelimbic cortex of control (Con) and maternally separated (MS) female rats at 25 (juvenile), 40 (adolescent), and 100 (adult) days of age. (**a**) The white arrows identify a double-labeled CB2/PV neuron; the yellow arrow indicates the PV-labeled neuron in Con subjects or (**b**) MS subjects; (**c**) a CB2 immunopositive neuron faintly stained in gray (i.e., DAB/nickel). Magnification is 20×, and the bar represents 100 µm. Stereological assessment of (**d**) PV neurons without additional labeling; means ± SE for n = 4/condition. * *p* < 0.05 for a main effect of age. (**e**) The total number of PV immunopositive neurons (PV single and CB2/PV counted cells). Both the main effect of age and the condition × age interaction were marginally significant (*p*’s = 0.05 and 0.06, respectively); and (**f**) counted cells immunoreactive for CB2 and PV double labeling across age and condition in MS females relative to controls. Means ± SE for n = 4/condition. * *p* < 0.05 indicates a significant interaction of Age × Condition; # *p* < 0.05: P25 vs. P100 in the MS group were significantly different. (**g**) Confocal imaging of CB2-immunoreactive neurons (green; 488 λ) and PV-immunoreactive neurons (red; 560 λ). The white arrows show CB2/PV interactions, with CB2-immunoreactive neurons wrapped around the PV-immunoreactive cell, indicating that these markers are not co-localized. The far-right picture illustrates the z-plane of the image.

**Figure 4 biomolecules-14-00464-f004:**
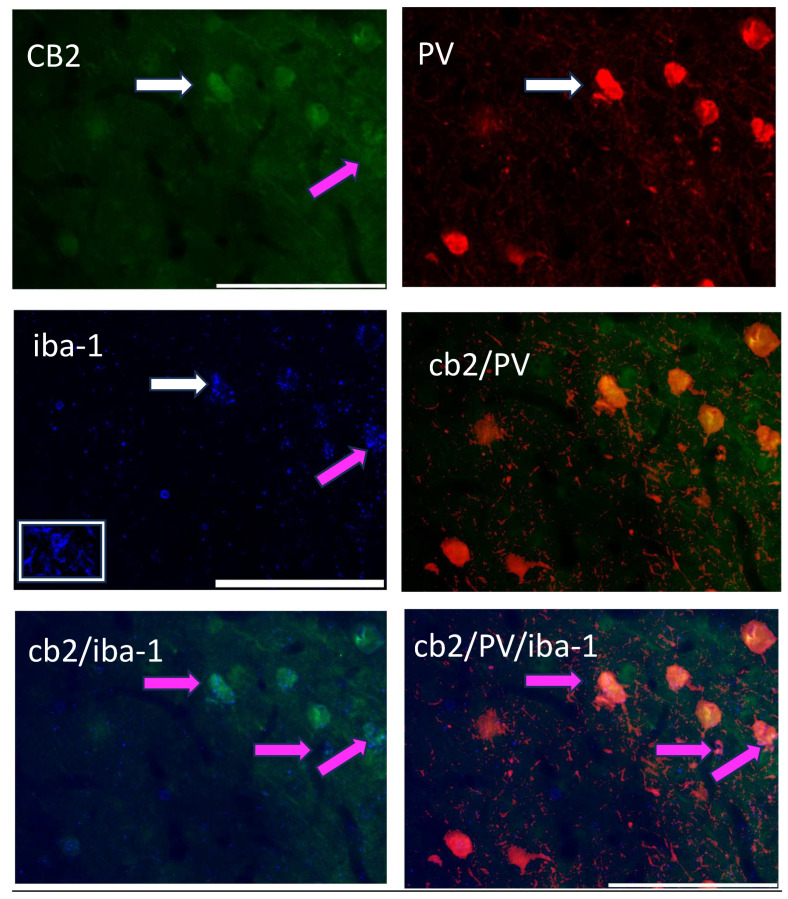
A triple-label image of CB2-immunoreactive cells (green; 488 λ; **upper left**), PV-immunoreactive neurons (red; 560 λ; **upper right**), the microglia marker, iba-1 (blue; 660 λ; **middle left**); this picture has an inset in the white box showing a microglia cell, CB2/PV image (**middle right**) as indicated in yellow; and their combined image (CB2/PV/iba-1; **lower right**). The white arrow points to the same cell in each image. Pink arrows and pink on the image indicate a CB2/iba-1 co-expression (**lower left and lower right**).

**Figure 5 biomolecules-14-00464-f005:**
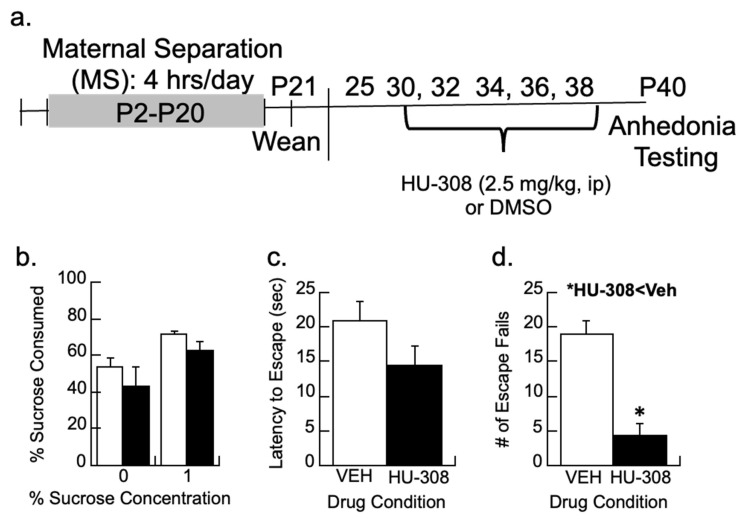
CB2 activity and depressive-like behaviors. (**a**) Timeline showing how maternally separated (MS) females were treated every other day with the CB2 agonist, HU-308 (2.5 mg/kg), or DMSO vehicle and tested for behavior. (**b**) Sucrose preference for a 1% sucrose solution was not significantly affected by HU-308. (**c**,**d**) While HU-308 did not significantly reduce the latency to escape, MS females treated with HU-308 had fewer escape fails than vehicle-treated females; * *p* < 0.05. Means ± SE for n = 8/group.

**Figure 6 biomolecules-14-00464-f006:**
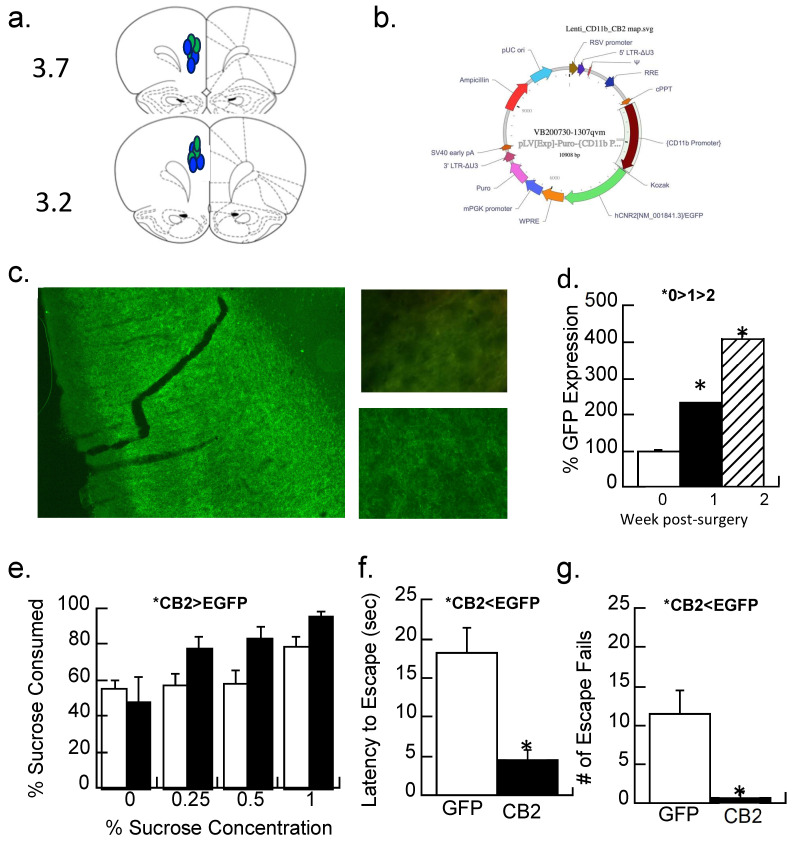
CB2 activity and depressive-like behaviors in maternally separated female rats at 40 days of age. (**a**,**b**) A CB2-EGFP or EGFP-alone expressing lentiviral vector with an EGFP reporter into the prelimbic cortex; (**a**) placements shown with blue indicating the CB2-EGFP virus, green the EGFP virus at the AP position of 3.7 or 3.2. (**b**) the cloning map used for plasmid construction for the CB2-EGFP virus. (**c**) Representative section showing the viral-mediated expression of the CB2-EGFP virus in the prelimbic cortex at 1.5×. Top picture shows background immunofluorescence in a control section compared with (bottom) CB2-EGFP expression one week following transduction; Image at 40×. (**d**) Quantitation of immunofluorescence 1 day after injection (0 weeks), 1 week, or 2 weeks after injection of the CB2-EGFP virus. Data are expressed as a percentage of the control injection at time 0. * *p* < 0.05; Means ± SE for n = 3/group. (**e**) MS females were given a two-bottle choice (water vs. 0, 0.25, 0.5, and 1% sucrose solutions) for two days each. Animals with CB2-EGFP overexpression drank more sucrose (less anhedonia) than EGFP-alone controls (* *p* < 0.05). Means ± SE for n = 8/group. (**f**,**g**) Prelimbic overexpression of the CB2 receptor significantly reduced the latency to escape and had fewer escape fails than subjects expressing the EGFP-alone virus; * *p* < 0.05. Means ± SE for n = 8/group.

## Data Availability

Data are available upon request.

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
