# Peer review of "Increasing CB2 Receptor Activity after Early Life Stress Prevents Depressive Behavior in Female Rats"

_biomolecules, 2024, doi:10.3390/biom14040464_

Round 1

Reviewer 1 Report

Comments and Suggestions for Authors

The paper „Treatment of the Progressive Loss of CB2 Receptors after Early Life Stress Prevents Depressive Behavior“ is well-designed and investigates a highly relevant topic. It is well structured and all necessary information is given. I truly enjoyed reading it. Overall, I only have minor comments.

General comments:

- Throughout the manuscript, the author sometimes refers to „we“. As there is only one author, this phrasing should be avoided.

Introduction:

The introduction would benefit from some background information on why adolescence is one period of interest for this study, especially as the author chose adolescence for several experiments. This information is given in the discussion, but I believe that providing this information in the introduction is beneficial for the reader, especially to those less familiar with the literature on adolescence.  

Methods:

General comment:

- were the two hemispheres analyzed separately? Or were results merged? Please provide this information for all analyses.

- Please specify why only females were included in this study and what might be the benefit of this design.

- 2.1. During what time of the day was MS performed?

- 2.3. Experiment 1.

Please specify why only adolescent animals were used for this analysis.

- 2.5.1. third line: it seems as if there is an „or“ too much in this sentence „…treated with the dimethyl sulfoxide vehicle (n=7) or the CB2 agonist, or HU-308 in dimethyl sulfoxide…“

Results

- It seems like the coloring of the bars in the figures is mixed up.

Author Response

The paper „Treatment of the Progressive Loss of CB2 Receptors after Early Life Stress Prevents Depressive Behavior“ is well-designed and investigates a highly relevant topic. It is well structured and all necessary information is given. I truly enjoyed reading it. Overall, I only have minor comments.

I sincerely thank you for your kind comment and the other astute comments listed below that strengthen the manuscript. I have addressed every comment as delineated below and highlight the changes in an annotated manuscript.

General comments:

- Throughout the manuscript, the author sometimes refers to „we“. As there is only one author, this phrasing should be avoided.

            I have rephrased the text to remove all personal pronouns from the manuscript. "I" did not sound right, even though I did the work.  I guess I am used to the royal "we." [Line 129, 541, 546,550, 555, 599, 606]

Introduction:

The introduction would benefit from some background information on why adolescence is one period of interest for this study, especially as the author chose adolescence for several experiments. This information is given in the discussion, but I believe that providing this information in the introduction is beneficial for the reader, especially to those less familiar with the literature on adolescence. 

I have now added the following information at the end of the Introduction. It reads [Lines 130-134]:

            "The focus on the trajectory between P25, 40, and 100 reflects childhood, adolescence, and adulthood in rats. More importantly, research shows that the depressive effects associated with adversity manifest during adolescence in humans and in rats [3]." 

Methods:

General comment:

- were the two hemispheres analyzed separately? Or were results merged? Please provide this information for all analyses.

            Section 2.4.1 now reads [Lines 232-234]:

             "Four serial coronal sections (intersection interval 240 µm)/animal of both hemispheres were analyzed, and the average value was used in subsequent calculations to represent each subject."

- Please specify why only females were included in this study and what might be the benefit of this design.

             I now include the following paragraph (same as above) [Lines 138-143]:

            "While both sexes demonstrate elevated inflammation, PV cell loss, and depressive behavior (Brenhouse et al., 2008; Lukkes et al., 2012), we focused exclusively on females in this study. First, the effects of a COX-2 inhibitor prevent PV loss in both sexes, but its ability to prevent depressive effects has only been demonstrated in females (Lukkes et al., 2012). Second, maternal separation increased microglia soma size and microglia arborization in females, but not in males (Gildawie et al., 2020)."

- 2.1. During what time of the day was MS performed?

   The following has been added [Lines 156-157]: "Separations occurred between the window of 7:00 AM-12:00 PM."

- 2.3. Experiment 1.Please specify why only adolescent animals were used for this analysis.

I have added the following to the Introduction [Lines 130-134]:

            "The focus on the trajectory between P25, 40, and 100 reflects childhood, adolescence, and adulthood in rats. More importantly, research shows that the depressive effects associated with adversity manifest during adolescence in humans and in rats [3]. "

- 2.5.1. third line: it seems as if there is an „or“ too much in this sentence „…treated with the dimethyl sulfoxide vehicle (n=7) or the CB2 agonist, or HU-308 in dimethyl sulfoxide…“

  Thank you.  It now reads [Lines 279-280]: "...rats that underwent maternal separation were treated with the dimethyl sulfoxide vehicle (DMSO; n=7) or the CB2 agonist, HU-308 in DMSO (2.5 mg/kg; n=6) on..."

Results

- It seems like the coloring of the bars in the figures is mixed up.

            My sincerest apologies! Something happened when I uploaded the figures. I will ensure that the results are as clear in the upload as they are when I submit them. 

Reviewer 2 Report

Comments and Suggestions for Authors

Susan Andersen investigated the possible involvement of the microglial CB2 receptor in animal models of depression. She found that early-life stress (maternal separation) caused a reduction in the expression of the CB2-associated enzyme (MARCH7) and an increase in its target (NLRP3). Immunohistochemistry assays indicated mixed findings - CB2-PV colocalization (Fig. 3a, Fig. 4), non-co-localization of CB2 and PV, or CB-Iba1-PV co-localization (Fig. 3c). Lastly, pharmacological activation of CB2 receptors by HU-308 or overexpression of CB2 (via AAV) in microglia of the prelimbic cortex failed to alter sucrose consumption but reduced the total number of escape failures and latency to escape in response to footshock stress in the maternally separated rats. She concluded that reduced microglial CB2 expression may underlie depression-like behavior in maternally separated rats. Overall, it is an interesting research topic, but the findings are conflicting and preliminary. The data presentation and explanation are confusing and not convincing. The presented data do not support the conclusion the author made.

Major concerns:

  1. There is no presented evidence indicating that the maternal separation (MS) rats displayed a decrease in microglial CB2 expression. MARCH7 and NLRP3 are not CB2-specific. Multiple factors can modulate their expression levels. To address this issue, double- or triple-staining IHC and ISH assays are required to determine whether CB2 is expressed in cortical microglia or neurons and whether its expression is lower in MS rats. Actually, growing evidence indicates CB2 receptor expression in neurons, not only in microglia.
  2. CB2-antibody: It is well-known that, so far, there is no specific CB2 antibody available. A justification for choosing a polyclonal CB2 antibody (Cayman) and more details about its epitope and signal/species specificity should be provided. An appropriate control (immune peptide or CB2-KO) should be included.
  3. The findings in Fig. 3a are conflicting and confusing. Individual CB2-staining, PV-staining, and the merged images should be provided in Fig. 3a to determine whether CB2 is co-localized with PV, and what percentage of PV neurons shows CB2 expression (quantitative cell counting data should be provided).
  4. Fig. 2b is confusing and difficult to understand. Why are the two bars the same color at 40- and 100-day groups? Which one is the control? What does the “# of CB2/PV neurons” mean? Does it show the percentage of PV neurons that express CB2? Does it show CB2 up-regulation at 25 days after maternal separation? Is this increase in CB2 expression statistically significant?
  5. Findings in Fig. 4 are also confusing. The images show CB2-PV-Iba1 co-localization. How do you know this co-localization is due to CB2 expression in microglia near PV neurons and not in PV neurons? Additional images with a high resolution of CB2-staining are required to show CB2-Iba1 co-localization in microglia, and quantitative cell count data should be provided.
  6. Fig. 5b: The author used the same color to label two bars (1% sucrose group). Which one is the control?
  7. Fig. 6: The validation images and cell count data should be provided to support AAV-mediated CB2 overexpression in microglia. The same color was used to label different groups in Fig. 6b. Is the difference between the groups significant?
Comments on the Quality of English Language

No concern about it

Author Response

Susan Andersen investigated the possible involvement of the microglial CB2 receptor in animal models of depression. She found that early-life stress (maternal separation) caused a reduction in the expression of the CB2-associated enzyme (MARCH7) and an increase in its target (NLRP3). Immunohistochemistry assays indicated mixed findings - CB2-PV colocalization (Fig. 3a, Fig. 4), non-co-localization of CB2 and PV, or CB-Iba1-PV co-localization (Fig. 3c). Lastly, pharmacological activation of CB2 receptors by HU-308 or overexpression of CB2 (via AAV) in microglia of the prelimbic cortex failed to alter sucrose consumption but reduced the total number of escape failures and latency to escape in response to footshock stress in the maternally separated rats. She concluded that reduced microglial CB2 expression may underlie depression-like behavior in maternally separated rats. Overall, it is an interesting research topic, but the findings are conflicting and preliminary. The data presentation and explanation are confusing and not convincing. The presented data do not support the conclusion the author made.

I sincerely thank you for your kind comment and the other astute comments listed below that strengthen the manuscript. I have addressed every comment as delineated below. An annotated copy is included.

            The current study examined the expression of CB2 receptors and their changes in MS animals. I am glad that you found the subject of interest. While the reviewer describes the results as "conflicting," they are presented in a manner that reflect a series of studies that led to the conclusion that CB2 receptors on microglia–and not PV neurons as originally hypothesized–are transiently upregulated and progressively lost with maturation as the same stages when depressive-like behaviors emerge in MS animals. It is not the perfect study. Rather, it is a series of experimentally solid assessments to support the conclusions and raise new and novel questions in the field of early adversity and inflammation. Indeed, more experiments can be conducted, but it does stand as an original contribution to the literature. It seems the other reviewers agree.

            Notably, Reviewer #2 brought up important criticisms. I added several additional pieces of information whenever possible. New data are delineated point-for-point below. Unfortunately, I cannot conduct further experiments at this time or access some of the data that was left behind [Western immunoblot]. However, several of the issues raised were already discussed in the text. For example, concerns about antibody specificity were in the discussion and maybe the reviewer missed this information. Other details are easily addressed by modifying the figure for improved clarification (e.g., the significance of the figure itself and not only indicated by a *p<0.05 in the figure caption and text). Viral expression of CB2 was by lentivirus, not AAV.

Here are my responses:

Major concerns:

1) There is no presented evidence indicating that the maternal separation (MS) rats displayed a decrease in microglial CB2 expression.

            I see your point; the title promises something that was not shown.  As a result, the new title is [Page 1]:

            Increasing CB2 Receptor Activity in Microglia after Early Life Stress Prevents Depressive Behavior in Female Rats

2) MARCH7 and NLRP3 are not CB2-specific. Multiple factors can modulate their expression levels.

            I agree. I understand that these inflammatory factors are specific for CB2. Rather, their inclusion advances their role as part of the well-established inflammatory effect of MS in the literature. To clarify, the introductory text now states [Lines 125-130]:

            "First, Western immunoblot was used to determine whether exposure to early adversity alters signaling pathways that are associated with inflammatory pathways, including those linked to CB2 receptors on microglia (e.g., the E3 ubiquitin enzyme MARCH7 and its target, [NLRP3] [50]."

Later on in the discussion, the text reads [Lines 514-516, 523-525]:

            "Notably, we found reduced MARCH7, a regulatory enzyme for NLRP3 activity, and an elevation in NLRP3, a critical factor in inflammasome activity, in the prelimbic cortex of maternally separated adolescent females. Together, these two indices provide a pathway that potentially links the CB2 receptor to enhanced cytokine production during early adversity. However, additional studies that directly manipulate the CB2 receptor on microglia are needed to confirm this relationship."

3) To address this issue, double- or triple-staining IHC and ISH assays are required to determine whether CB2 is expressed in cortical microglia or neurons and whether its expression is lower in MS rats. Actually, growing evidence indicates CB2 receptor expression in neurons, not only in microglia.

            The current results did use double and triple staining IHC and determined that CB2 is not on PV neurons, although their expression is highly intertwined (Fig 3). Fig 4 shows co-localization with iba-1. I have included an additional figure in the panel to show this more clearly.

4) CB2-antibody: It is well-known that, so far, there is no specific CB2 antibody available. A justification for choosing a polyclonal CB2 antibody (Cayman) and more details about its epitope and signal/species specificity should be provided. An appropriate control (immune peptide or CB2-KO) should be included.

The text now reads [Lines 606-624]:

            "However, caution is needed in interpreting the current results. Concern over commercially available CB2 antibodies has hampered research on this molecule [86]. The Cayman antibody binds to the intracellular 3rd loop (residues 228–242; [86]) of the N-terminus [87]. Testing of the Cayman antibody used in the current study has revealed high sensitivity to the CB2 receptor, as indicated by positive controls using mass spectrometry and blocking Western immunoblot expression with a CB2 antibody [88]. Dependence on mice knock-outs to characterize CB2 antibodies is potentially flawed given different epitopes for antibody binding and knock-out sites [86]. Until a full CB2 receptor knock-out is available, relying on partial knock-outs found in the Deltagen and Zimmer mice strain will continue. The Cayman antibody also binds to other proteins, but whether they are different epitopes of the CB2 receptor is still unclear [88]. Specificity for the CB2 receptor was reported to be at least 50-70%. Yet, the inflammatory signaling pathways and the behavioral studies are consistent with changes in CB2 activity following maternal separation. Most importantly, the behavioral effects of the CB2 agonist HU-308 are confirmed with the specific lentiviral vector expressing CB2-EGFP on microglia. Immunofluorescence of the reporter gene indicates viral overexpression is 2.3-fold higher than controls when the animals were behaviorally tested. However, additional confirmation with specific mRNA transcripts is needed and will provide greater insight into the degree of CB2 elevation found in maternally separated females."

5) The findings in Fig. 3a are conflicting and confusing. Individual CB2-staining, PV-staining, and the merged images should be provided in Fig. 3a to determine whether CB2 is co-localized with PV, and what percentage of PV neurons shows CB2 expression (quantitative cell counting data should be provided).

            Fig. 3 shows individual CB2 staining, which is faint and gray in the inset and therefore was too unreliable to quantify on its own. I now provided a larger picture of the staining of the individual CB2 staining in Fig. 3c. PV alone staining is also highlighted by the yellow arrows, and the two co-stained neurons are highlighted with the white arrow. Quantitative cell counting data are provided in Fig 3d-f. Methodologically, if the sections were allowed to further develop to allow the CB2-nickel to show up better, it would have been impossible to determine the PV and CB2 immunoreactivity as the PV neurons would be too dark.

6) Fig. 2b is confusing and difficult to understand. Why are the two bars the same color at 40- and 100-day groups? Which one is the control? What does the “# of CB2/PV neurons” mean? Does it show the percentage of PV neurons that express CB2? Does it show CB2 up-regulation at 25 days after maternal separation? Is this increase in CB2 expression statistically significant?

            I think the reviewer is referring to Fig. 3, which illustrates the above-mentioned data, not Fig. 2.  I sincerely apologize for the poor upload of the figures and have addressed it in the revision to highlight the two groups: Con in white bars and MS in black bars. I have also included additional statistical analyses and the text now reads [Line 385-396]:

            " In contrast to our previous studies in male rats [17, 29], a significant main effect demonstrated increased PV immunoreactive cells in maternally separated females relative to controls (F1, 17 = 4.7, p<0.05), with no significant interaction (Figure 3d). The total amount of PV-immunoreactive cells was marginally significant (p=0.05; Figure 3e) as was the interaction by age (p=0.06). Double-labeled neurons of CB2/PV were counted (Figure 3f). While CB2/PV did not appreciably change with age in control subjects, CB2/PV declined across age in the maternally separated animals (Age x Condition interaction: F2,17 = 4.20, p<0.05). When the interaction was dissected further with post-hoc ANOVA, it was revealed that CB2/PV  neurons significantly changed across age in the MS group, with significant differences observed between P25 and P100 (p<0.05) but counts were not different between P25 and P40 (p=0.07). CB2/PV immunoreactivity did not change significantly across age in the Con group."

7) Findings in Fig. 4 are also confusing. The images show CB2-PV-Iba1 co-localization. How do you know this co-localization is due to CB2 expression in microglia near PV neurons and not in PV neurons? Additional images with a high resolution of CB2-staining are required to show CB2-Iba1 co-localization in microglia, and quantitative cell count data should be provided.

            Given the findings with specific viral overexpression of CB2 on microglia (Fig 6) and the lack of co-localization in Fig 3g, the observation that CB2 and iba-1 are co-expressed is unclear with that staining approach. In Figure 4, I have added another image with the PV subtracted out, showing CB2 and iba-1 co-expression with the pink arrows. However, I cannot perform any quantitative cell counts at this time as I no longer have a laboratory (and the samples).

8) Fig. 5b: The author used the same color to label two bars (1% sucrose group). Which one is the control?

            Uploading issue that has now been corrected.

9) Fig. 6: The validation images and cell count data should be provided to support AAV-mediated CB2 overexpression in microglia.

            Agreed! The lentiviral vector has an EGFP reporter as shown in its map in Fig 6b. I now include images showing EFGP expression in transduced animals within the prelimbic cortex at 1.5x and 40x, where fluorescent processes are abundant relative to background immunofluorescence (Fig 6c). The soma does not express the virus clearly, consistent with what others have shown (Sergijenko et al., 2013). Therefore, ImageJ was used to quantify the amount of fluorescence emitted from the EGFP reporter at three time points: 1 day after injection (e.g., week 0) and 1 and 2 weeks after injection. These data show that expression was a 2.3-fold increase from baseline when the animals were tested 1-week post-surgery.

10) The same color was used to label different groups in Fig. 6b. Is the difference between the groups significant?

            Uploading issue that has now been corrected. The significant results are now stated above the figure. They are also discussed in both the text and mentioned in the figure caption. I hope this is clear now.

Reviewer 3 Report

Comments and Suggestions for Authors

In the reviewed manuscript entitled ‘Treatment of the Progressive Loss of CB2 Receptors after Early Life Stress Prevents Depressive Behavior’, the author assesses how maternal separation interacts with the endocannabinoid system to impact the expression of depressive-like behavior in the female offspring. This is an important issue due to the increasing number of people with the problem of mental disorders including depression. Before being accepted for publication, it is worth considering some additions and correcting editorial mistakes.

1. Why are studies limited to assessing changes only in female offspring? There is no justification for this limitation in the manuscript.

2. The title of the publication should include information that the research concerns female rats.

3. Section 2.5.1. In the sentence: Lower doses of 0.5 and 1 mg/kg HU-398 were ineffective in animal models of seizures [56]. It should be HU-308.

4. Please modify the presentation of results to make it more readable:

- add points for each result on the bars

- modify the color/pattern of the control bars - for some they are displayed almost completely black, as for other groups.

5. Figure 3b. What does the asterisk next to the legend refer to?

6. Section 3.2.3 In the sentence However, when this image is placed in the context of Figure 3d, the CB2/iba-1 are likely co-localized and wrapped around the PV neurons, as suggested by… There is no Figure 3d in the manuscript

7. Please expand the PFC abbreviation at the end of the discussion or replace it with the full structure name.

8. Why is a Western immunoblot of MARCH 7 available in the supplement but the original image for NLRP3 is missing? Please add them.

Author Response

I sincerely thank you for your astute comments listed below that strengthen the manuscript. I have addressed every comment as delineated below. An annotated copy is included.

In the reviewed manuscript entitled ‘Treatment of the Progressive Loss of CB2 Receptors after Early Life Stress Prevents Depressive Behavior’, the author assesses how maternal separation interacts with the endocannabinoid system to impact the expression of depressive-like behavior in the female offspring. This is an important issue due to the increasing number of people with the problem of mental disorders including depression. Before being accepted for publication, it is worth considering some additions and correcting editorial mistakes.

  1. Why are studies limited to assessing changes only in female offspring? There is no justification for this limitation in the manuscript.

I have now added the following text to clarify the use of females only [Lines 138-142]:

            "While both sexes demonstrate elevated inflammation, PV cell loss, and depressive behavior [17,31], we focused exclusively on females in this study. First, the effects of a COX-2 inhibitor prevent PV loss in both sexes, but its ability to prevent depressive effects has only been demonstrated in females [31]. Second, maternal separation increased microglia soma size and microglia arborization in females, but not in males [48]."

  1. ​The title of the publication should include information that the research concerns female rats.

The article is now titled:

            "Increasing CB2 Receptor Activity in Microglia after Early Life Stress Prevents Depressive Behavior in Female Rats"

  1. Section 2.5.1. In the sentence: Lower doses of 0.5 and 1 mg/kg HU-398 were ineffective in animal models of seizures [56]. It should be HU-308.

            Thank you and it is corrected [Line 285].

  1. Please modify the presentation of results to make it more readable:

            I have modified the majority of the figures for clarity. Most of the issues were due to an uploading error that is now fixed.

  1. Figure 3b. What does the asterisk next to the legend refer to?

            I apologize for the confusion. It was intended to show group differences as stated in the Figure caption. However, I revised the figure and now the overall significant effects are included in the figure itself. For example, Fig 3d has "*MS>Con" to show the overall main effect. The asterisk is then defined in the caption as *p<0.05.

  1. Section 3.2.3 In the sentence "However, when this image is placed in the context of Figure 3d, the CB2/iba-1 are likely co-localized and wrapped around the PV neurons, as suggested by… "There is no Figure 3d in the manuscript

            You are corrrect! I changed it to Figure 4 and have now changed it accordingly in the text to read [Line 434]:

            "However, when this image is placed in the context of Figure 4,..." I apologize for the misdirection.

  1. Please expand the PFC abbreviation at the end of the discussion or replace it with the full structure name.

            Done! [Line 648]

  1. Why is a Western immunoblot of MARCH 7 available in the supplement but the original image for NLRP3 is missing? Please add them.

            I would if I could.  I closed my laboratory 2 years ago and those images are no longer available.  

Round 2

Reviewer 2 Report

Comments and Suggestions for Authors

The author has appropriately addressed most of my concerns. I have no additional comments on the revision of the manuscript. 

Author Response

March 27, 2024

Revision: biomolecules-2842664: "Increasing CB2 Receptor Activity after Early Life Stress Prevents Depressive Behavior in Female Rats"

Dear Editor and Reviewer #2,

Words cannot express how grateful I am that you are both willing to work with me to find a way to get my data published.  We all agree that there are some interesting and important findings here, and other circumstances where I would have an active lab, I would be more than willing to address the very valid comments head-on.

With that in mind, I have clarified the model and worked on the wording of the manuscript to be as clear and explicitly correct in what the study is and is not. I hope you agree. My specific corrections are delineated below.

Clarifying the model. The model and the data supporting the model need additional clarifications.

1A. My understanding of the model is that MS leads to a reduction in CB2 signaling which in turn lead to reduced MARCH7 and an increase in NLRP3 activity leading to increase inflammation in the prelimbic area. If this is correct, please make this outline clearer in fig 2a figure legend.

  • The Fig 2a legend has been amended to now read (Lines 361-376):

                  " Figure 2: Figure 2: Inflammatory pathways disrupted by maternal separation (MS). a) Schematic showing signaling pathways associated with inflammation. Most research focuses on the TLR pathway. CB2 receptor-associated pathways also modulate MARCH7/axotropin and NLRP3 as shown here. The gray box insert shows the observed changes following MS, where CB2 overexpression leads to a down-regulation followed by disinhibition of NLRP3. Modified figure from DOI: 10.1097/HRP.0000000000000325). b, c) Female rats underwent MS or were controls (Con). At P40, the prelimbic prefrontal cortex was analyzed with Western immunoblots for MARCH7 and NLRP3 with VCP or actin as controls, respectively. b) MS rats have a significant decrease in the regulatory enzyme MARCH7. c) NLRP3 expression increases in MS females relative to controls (*p<0.05). Means ± SE are presented (n=6 /5 for MARCH7 and 5/5 for NLRP3). A second cohort replicated this finding."

1B. Except for the data shown in Fig 2b and 2c, I see no other data that supports the claim that MS reduces CB2 signaling in the prelimbic area. I recognize that the numbers of CB2/PV cells are trending down in MS from P25-100 and that this is not the case with Control, but this is because the levels at P25 are very high. In fact, even at P40 the levels seem higher (though not sig) in MS (Fig 3F). This is an important point because it raises questions about the rationale for using the CB2 agonist and overexpressing CB2 receptors. Please clarify if CB2 signaling is increased or is reduced in MS and what is the data to support this assertion.

  • I appreciate the reviewer's point. The focus should be on the P25 increase in CB2/PV counts, which I hypothesize is a protective mechanism triggered by MS. This hypothesis is consistent with other reports of elevated CB2 signaling after trauma or hypoxia, as cited in lines 117-118. The progressive loss of CB2 signaling by P40 (and P100) results in the loss of "protection." The reinstatement of protection therefore is the rationale for treatment with the CB2 agonist or the CB2 virus. The third paragraph in the discussion explores this hypothesis. I have tried to make this point clearer in other parts of the text, as evidenced by changes in lines 534-542.

1C. Please note that NLRP3 is expressed in other cells besides microglia. Most notably endothelial cells, astrocytes (under some conditions) and some neurons. I would therefore be very careful about stating that microglia are activated in your system or that there is evidence for inflammation. Similarly, is there data showing elevated IL1b in the prelimbic area in P40 females? Please clarify.

  • I have clarified that NLRP3 is found in other cell types as the reviewer states (Lines 130 and again 523).

            "NLRP3 is found on other cells, including epithelial cells, astrocytes and some neuronal populations {Ghaffaripour Jahromi, 2024 #17515}. While the data are consistent with the proposed model in Figure 2, additional studies that directly manipulate the CB2 receptor on microglia and other cell types are needed to confirm this relationship with MARCH7/NLRP3."

  • I have also added the following text discussing what is known about IL-1beta in MS animals. Lines 111-113 read:

            "Elevated levels of IL-1 b following MS in females are found {Ye, 2019 #17514}, but not consistently (Grassi-Oliveira, 2016 #16154}."

  1. Transfecting microglia and the role that microglia play in modifying behavior. The data presented in Fig 6 c-d confirms an increased expression of the CB2 receptor, but it does not conclusively demonstrate that microglia were transfected and expressing CB2R-GFP. This needs to be stated clearly and guide the interpretation of the data and the discussion. Even with these limitations, this experiment is important because it demonstrates that overexpression of CB2 in the prelimbic area is sufficient to reverse depression-like behavior in MS females. However, additional work is needed to clarify the exact cell types transfected and the mechanisms responsible for these behavioral changes.
  • Understood and I agree. The key picture I had to address this issue was not available. Unfortunate but here is how I addressed this point (Lines xxxx):

                  " While the intended target was microglia via the CD11b promotor, in the absence of immunohistochemistry directly localizing CB2 overexpression to microglia, the manipulation will be identified as CB2-EGFP overexpression only."

  • Later in the discussion, the following was modified and text added (Lines 305-308):

            "The effects of the CB2-EGFP virus suggest that introducing it into the prelimbic cortex was sufficient to increase CB2 expression following minimal physical trauma due to its delivery. However, localization to microglia was not confirmed and therefore additional studies will be needed to determine the cell-specificity of the observed behavioral effects."

  1. Colocalization of CB2 receptors on microglia. The colocalization between Iba1, CB2 and PV (Fig 4) has improved in the revised manuscript. However, based on the data, it is impossible to determine whether the CB2 receptor is expressed on microglia and/or on PV/CB2 positive cells that are being "hugged" and pruned by microglia. This limitation and the different interpretations of the data need to be clarified.
  • It is an interesting situation. I agree and have addressed it accordingly in paragraph four of the discussion (Lines 544-558):

  1. More cautious interpretation of the data and some rewording suggestions.

4A. Results exp 3.1. “To determine whether the effects of CB2 loss in maternally separated female rats were preventable, subjects were treated with the CB2 agonist 2.5 mg/kg HU-308 on P30, 32, 34, 36, and 38 following our previous intervention protocol (timeline in Figure 5a; [17]). The treatment was partially effective in reducing the depressive-like behaviors of anhedonia by P40 as assessed by escape behavior and sucrose preferences (Figures 5b-d).”

I suggest rewording, “To determine whether activation of CB2 signaling could reverse the depression-like behavior in MS females, we…”

  • Thank you. Line 452: The change has been made, although I prefer "prevent" instead of "reverse" as previous observations from my lab indicate no depressive behavior before ~P40.

4B. Discussion p20: “The results of the current studies further implicate the activation of microglial processes following exposure to early adversity, although additional studies are needed”.

I do not see convincing evidence for microglial activation. Please clarify or restate.

  • Understood. The text was modified and now reads (Lines 508-510):

            "The results of the current studies demonstrate a role for the CB2 receptor in depressive-like behavior in adolescent female rats following exposure to early adversity; additional studies are needed to show localization to microglia conclusively."

4C. Discussion p20. “Increases in CB2 receptor-associated signaling molecules ultimately linked to the elevation of cytokine production are altered in maternally separated females.”

I find this confusing; wouldn’t you expect that increased CB2 signaling would reduce inflammation rather than increase it?  I thought the model is that MS reduces CB2 signaling…Also, what is the evidence that MS reduces CB2 signaling (see above). Please clarify.

  • I see your point. I have modified the above sentence to read more accurately (Lines 510-512):

            "Initial increases in CB2 receptor-associated signaling molecules followed by their compensatory changes provide a possible mechanism for the elevation of cytokine production in maternally separated females."

  • I apologize if the MS and CB2 model is not clear. In addition to modifying the model in Figure 2, I have added the following to the text to clarify Lines 537-543:

            " The initial increase in CB2 expression associated with PV neurons at P25 likely represents a compensatory mechanism of primed that fails over time. By P40, CB2-associated rescue of these PV neurons (i.e., CB2/PV expression) is insufficient to prevent depressive behavior. Treatment with a CB2 agonist before P40 reduced depressive behavior (escape fails but not sucrose preferences). Moreover, the CB2-EGFP lentiviral vector confirmed the role of overexpression of CB2 in depressive effects (both escape fails and enhanced sucrose preferences)."

4D. top of P21. “Moreover, the lentiviral vector confirmed the role of overexpression of CB2 on microglia in depressive effects.”

There is no evidence that this virus targeted microglia. Please clarify or restate.

  • Agreed, and already addressed per #2. Here are the changes (Lines 305-308):

                  " While the intended target was microglia via the CD11b promotor, in the absence of immunohistochemistry directly localizing CB2 overexpression to microglia, the manipulation will be identified as CB2-EGFP overexpression only."

  • Later in the discussion, the following was modified and text added (Lines 561-564):

            "The effects of the CB2-EGFP virus suggest that introducing it into the prelimbic cortex was sufficient to increase CB2 expression following minimal physical trauma due to its delivery. However, localization to microglia was not confirmed and therefore additional studies will be needed to determine the cell-specificity of the observed behavioral effects."

  • And more (Lines 541-543): "Moreover, the CB2-EGFP lentiviral vector confirmed the role of overexpression of CB2 in depressive effects (both escape fails and enhanced sucrose preferences)."

4E. top of p21. “As a preliminary assessment to probe how CB2 selectively modulates activated microglia... PV immunoreactivity in juvenile animals [17].”

This entire paragraph needs rewording and clarifications. CD11b is not a good marker of “activated microglia” and there is no data to suggest that the virus targeted microglia or that CB2 is expressed on microglia. The paragraph implies that increased CB2 signaling enhances microglial activation and inflammation which is confusing because I thought CB2 signaling reduces NLRP3 and inflammation. Please clarify.

  • I have re-written this first part of the discussion. Rather than send the reviewer to specific lines, I kindly refer the reviewer to the revised discussion.

4F. Page 21. Future studies will use FRET to examine the amount of PV expression within each cell. FRET is used to assess protein-protein interaction. Do you mean that you will use fluorescence intensity to quantify PV expression?

  • Indeed, you are correct. The passage now reads (Line 582):

            "Future studies will determine the density of the staining (or immunofluorescence if these secondary antibodies are used) to examine the amount of PV expression within each cell."

4G. Middle of page 23. “Most importantly, the behavioral effects of the CB2 agonist HU-308 are confirmed with the specific lentiviral vector expressing CB2-EGFP on microglia.  (Line 631).

See above concerns regarding the targeting of the virus.

  • As per the earlier comments, I have gone through the manuscript to no longer state that the virus was microglia specific.

4H. Bottom page 23. “The effects of the CB2-EGFP virus suggest that introducing it into the prelimbic cortex was sufficient to increase microglial activity (and presumably CB2 expression) following minimal physical trauma due to its delivery."

                  There is no evidence that the virus targeted microglia or altered their activity. Please clarify or reword.

  • This statement now reads (Lines 643-647):

            "The effects of the CB2-EGFP virus suggest that introducing it into the prelimbic cortex was sufficient to increase CB2 expression following minimal physical trauma due to its delivery. However, localization to microglia was not confirmed and therefore additional studies will be needed to determine the cell-specificity of the observed behavioral effects."

4I. Conclusion. Page 24. “The current studies support the protective role of CB2 receptors on microglia in maternally separated females.”

  • This statement now reads (Lines 675-676):

"The current studies support the protective role of CB2 receptors in the prelimbic cortex of maternally separated females."

  1. Figures

Fig 2A- Please add couple of sentences to clarify the model (e.g., CB2 activates March7 and March7 downregulates NLRP3)

  • I have revised the model as shown in Figure 2 to help clarify the issue.

Fig 2B Gel- there is an extra lane that does not make sense. Also, please label lanes as Control and MS.

  • The label for the gels was stated above, with the extra lane in 1 serving as the negative control. Lanes B1-6 are control, Y1-5 are MS animals, an empty lane and then the marker.

Fig 2C gel- please use uniform sized lanes, each square is a different size.

  • Yes-fixed.

Fig 5A- Please reposition days P25 and P30 so they are not on top of one another.

  • Done.

Again, I would like to sincerely thank the reviewer for their thoughtful comments and time spent on improving this manuscript. Clearly, the manuscript is greatly improved, and notably a guide for future research efforts.

Sincerely,

Sue Andersen, Ph.D.